# Geographic and Socioeconomic Influence on Knowledge and Practices Related to Antimicrobial Resistance among Smallholder Pig Farmers in Uganda

**DOI:** 10.3390/antibiotics11020251

**Published:** 2022-02-15

**Authors:** Sandra Nohrborg, Michel Mainack Dione, Amia Christine Winfred, Louis Okello, Barbara Wieland, Ulf Magnusson

**Affiliations:** 1Department of Clinical Sciences, Swedish University of Agricultural Sciences, SE-750 07 Uppsala, Sweden; sandra.nohrborg@slu.se; 2Animal and Human Health Program, International Livestock Research Institute, Ouakam, Dakar BP 24265, Senegal; m.dione@cgiar.org; 3Mott MacDonald Uganda Limited, Kampala 10101, Uganda; winfred.amia@mottmac.com; 4Independent Researcher, Kampala 10101, Uganda; louisomoya@gmail.com; 5Institute of Virology and Immunology, CH-3147 Mittelhausern, Switzerland; barbara.wieland@ivi.admin.ch; 6Department of Infectious Diseases and Pathobiology, Vetsuisse Faculty, University of Bern, CH-3012 Bern, Switzerland

**Keywords:** antimicrobial use, antimicrobial resistance, smallholders, pig farming, livestock, farm practices, knowledge, access

## Abstract

To mitigate the development of antimicrobial resistance (AMR), antibiotic use (ABU) in the livestock sector needs to be reduced. In low- and middle-income countries, regulations have shown to be less successful in reducing ABU. Here, a bottom-up approach can complement legal frameworks, which requires an understanding of the drivers for ABU. In this study, we investigated the influence of geographic and socioeconomic settings on determinants for ABU among pig farmers in Uganda. The data were collected through a questionnaire in two districts, Lira and Mukono, and comparative statistical analyses were performed. Farmers in Lira had less access to animal health services, applied disease prevention measures less and used antibiotics more. In Mukono, it was more common to consult a veterinarian in response to disease, while in Lira it was more common to consult an animal health worker. There was no difference in how many farmers followed treatment instructions from a veterinarian, but it was more common in Lira to follow instructions from pharmacies. The findings support the need for locally tailored AMR-reducing interventions to complement regulations. To accomplish this tailoring, systematic collection of knowledge of farm structures, farm practices and access to animal health services and veterinary drugs is necessary.

## 1. Introduction

Antimicrobials have been, and still are, invaluable tools in treating bacterial diseases in both humans and animals. Therefore, the development of antimicrobial resistance (AMR) is a major threat to public and animal health [1,2]. Besides the direct health consequences, it may also threaten food security and livelihoods, especially in low- and middle-income countries (LMICs) [3].

Even though AMR is a naturally occurring mechanism in bacteria [4,5,6], the widespread use of antibiotics in humans and livestock has accelerated this development. Consequently, AMR has reached alarming levels in many parts of the world [7,8,9,10]. Thus, a future without access to effective antibiotic treatment might be a reality if this development is not mitigated. Since bacteria can spread between animals and humans [11], either directly or indirectly (e.g., through food products or the environment), AMR is a true ‘One Health’ issue that needs to be handled at the global level together by the human and livestock sectors, including the biosciences and social sciences, in an interdisciplinary manner [12].

It has been estimated that the global antibiotic use in the livestock sector exceeds their use in humans [8]. Therefore, it is essential to stop over- and misuse of antibiotics in the livestock sector. Antibiotics in livestock production are used not only for treatment of disease but also in more imprudent ways, such as for disease prevention (prophylaxis) and growth promotion [1,13]. Other problems are the use of antibiotics without proper diagnostics or lack of access to the best-suited antibiotics. Extensive use of antibiotics is also facilitated by weak legislation and guidelines on antibiotic sales and use, and in some settings insufficient enforcement of, and compliance with, such regulations [13,14,15].

Studies have indicated that AMR levels in bacteria isolated from livestock have increased in LMICs since the beginning of the millennium, with the largest increase in poultry and pig production [10]. However, the level of AMR differs between regions, and so far sub-Saharan Africa (SSA) seems to have a lower level of AMR than, for example, Southeast Asia. Current and emerging hotspots for AMR bacteria in livestock have, however, been identified in Egypt, South Africa and Kenya, indicating that the problem is existent and on the rise in Africa as well [10]. In Uganda, from which the data presented in this paper are collected, studies have shown varying frequency of AMR bacteria, including multi-drug resistant ones, in both human and animal samples [13,16,17,18,19,20,21,22]. Neighboring countries, such as Kenya and Tanzania, seem to share a similar situation, according to several studies [23,24,25]. However, the overall AMR burden in SSA is difficult to evaluate because of the limited number of studies conducted in this region compared to other LMIC regions, such as South and Southeast Asia [10].

In LMICs, such as Uganda, where small-scale production is still dominant and with a generally low awareness about AMR among the farmers, legislation and other regulations have been shown to be less successful in reducing antibiotic use [14,15]. Limited resources to enforce the regulations [13] and/or poor infrastructure in rural areas can make compliance control difficult. In addition, poor access to professional animal health services and insufficient laboratory capacity can compromise the farmers’ abilities to follow regulations and to perform the ‘best practices’ from an AMR perspective [26].

Therefore, a more bottom-up approach showing the farmers how to reduce the need for antibiotics and use them in a medically rational way may be a promising approach to complement existing legal frameworks. However, this requires an understanding of the drivers for antibiotic use by the farmers. By identifying and understanding these drivers, interventions can be tailored to fit different socioeconomic circumstances [27,28]. Thus, farmers’ access to veterinary advice and drugs, as well as their knowledge and practices related to AMR, are critical parameters to evaluate.

The overall objective of this study was to increase our understanding about how to improve the tailoring of interventions aimed to refine smallholders’ use of antimicrobials in livestock in LMICs. Specifically, we wanted to investigate the possible influence of geographic and socioeconomic settings on some major determinants for antibiotic use among pig farmers within the same regulatory framework. Thus, we designed a study targeting two districts in Uganda with different production characteristics and compared the smallholder pig farmers’ access to animal health services and veterinary drugs, and their practices and knowledge related to antibiotics and AMR. In short, we found that legal regulations alone are not sufficient to control antimicrobial use in livestock.

## 2. Results

### 2.1. Background Information

The demographics and socioeconomic characteristics of the farmers, farm characteristics, feeding and manure handling routines and pig disease occurrence in the two districts are presented in detail in Appendix A. These data support the already perceived view of Lira being a more rural district where people to a higher degree depend on livestock than in peri-urban Mukono. For example, livestock contributed more to the household income in Lira than in Mukono and it was more common in Mukono to have the main income source outside the farm. We also identified signs of a more advanced livestock production in Mukono, with hired workers to a larger extent, more pigs per farm and keeping the pigs housed instead of tethered. Selling live pigs and feeding the pigs with commercial or pre-mixed feed or restaurant/household waste was also more common in Mukono, which might indicate better market access in the district, being more urban than Lira, and given its proximity to the capital Kampala. In Lira, on the other hand, scavenging and pasture grazing were common feeding practices, further reflecting the more rural and traditional farm conditions in the district. Also, farmers in Mukono to a larger extent made use of the pig manure as a fertilizer while farmers in Lira more often discarded or left the manure in open air. Farmers in Lira were also challenged with higher disease rates among their pigs.

### 2.2. Access to Animal Health Services and Veterinary Drugs

The data on access to, and use of, animal health services and veterinary drugs at the investigated farms are presented in Table 1.

The access to professional animal health services differed significantly between the districts (*p* = 0.000), where 59% of the farms in Lira and 94% of the farms in Mukono had access to such services. Of those who had access to animal health services, the most common animal health service that could be accessed in both districts was a private full-time animal health worker, but the proportion was higher in Lira (*p* = 0.000). Access to both private and state or governmental animal health services was in turn more common in Mukono than in Lira (*p* = 0.001). In both districts, it was almost non-existent that the animal health services included laboratory testing. Further, it was more common that the farms in Lira were involved in a regular animal health service program, such as a vaccination program, compared to the farms in Mukono (*p* = 0.000).

No difference between the districts regarding access to pharmaceuticals or veterinary drugs could be shown, with about two-thirds of the farms in each district having such access.

### 2.3. Disease Prevention

The data on disease prevention measures at the investigated farms are presented in Table 2.

In Mukono, 73% of the respondents said that they had particular means to protect their animals from disease (all species included), which was more common than in Lira (*p* = 0.000). In both districts, the most common disease prevention measures for pigs were the use of veterinary drugs (including use of vaccines) and not mixing the pigs with other herds or flocks of animals.

### 2.4. Response to Disease

The data on responses to pig disease problems at the investigated farms are presented in Table 3.

In Lira, the most common responses to disease issues in pigs were to do nothing (35% of the respondents) or to consult a community animal health worker (34%). Both these proportions were higher in Lira than in Mukono (*p* = 0.000 and *p* = 0.000). However, consulting a private veterinarian was more common in Mukono than in Lira (*p* = 0.000). In both districts, it was less common than previously mentioned responses to consult a governmental veterinarian or to use medicine from a veterinary drug store.

### 2.5. Veterinary Drug Use Routines

The data on routines regarding veterinary drug use at the investigated farms are presented in Table 4.

Among those who had access to pharmaceuticals or veterinary drugs, there was no detectable difference between the districts in how many followed treatment instructions from a veterinarian (about half of the respondents in both districts). However, following instructions from an animal health worker was more common in Mukono than in Lira (*p* = 0.0053), and, in contrast, following instructions from pharmacies was more common in Lira than in Mukono (*p* = 0.0005).

### 2.6. Antibiotic Use

The data on antibiotic use at the investigated farms are presented in Table 5.

Regarding antibiotic use in pigs during the four-week period prior to the study, a higher proportion of farmers in Lira had used antibiotics compared to farmers in Mukono (*p* = 0.001), with almost one in five in Lira and almost one in ten in Mukono. However, no significant difference could be found when it came to the proportion of farmers that had assigned an antibiotic as their most commonly used drug during this four-week period.

### 2.7. Effect and Handling of Veterinary Drugs

The data on effect and handling of veterinary drugs at the investigated farms are presented in Table 6.

It was about equally common in both districts that respondents had experienced that drugs did not work. Expired veterinary drugs were most commonly disposed of in Lira, as well as in Mukono. To return expired drugs to a pharmacy, give to another farmer or to use for the intended treatment did not occur on the farms in Lira and was almost non-existent in Mukono. Notable also was that almost half of the respondents in both districts said that they never had experienced expired drugs.

### 2.8. Knowledge about Vaccines and Antibiotics

The data on the respondents’ knowledge about what vaccines and antibiotics do are presented in Table 7.

The belief that vaccinations prevent animals from becoming sick was clearly the most common in both districts. However, there was a significant difference between the districts, where this proportion was higher in Mukono than in Lira. There was also a higher proportion of respondents in Lira than in Mukono that believed that vaccinations both cure and prevent animals from becoming sick.

Regarding the question about what antibiotics do, the proportion of respondents that believed that antibiotics cure sick animals was higher in Lira than in Mukono. The same was true for the proportion of respondents that believed that antibiotics prevent animals from becoming sick. However, the proportion of respondents that believed that antibiotics both cure sick animals and prevent them from becoming sick was higher in Mukono than in Lira. Further, in both districts, the proportions of respondents that believed that antibiotics are used for fattening were low.

## 3. Discussion

Our survey data highlight important differences between the districts, where farmers in Lira, the more rural of the two districts, were challenged with higher disease rates among their pigs, applied fewer biosecurity measures, had lower access to professional animal health services and used antibiotics more frequently with the instructions on veterinary drug use obtained more often from pharmacies than in Mukono.

From a disease, antibiotic use and AMR development perspective, several of the analyzed farm characteristics can play important roles, such as number of pigs kept, housing system, number of animal species kept, use of hired workers and selling of live animals. Differences in such farm characteristics and socioeconomic factors between the investigated districts result in different challenges to reduce antibiotic use and imply the need for tailored interventions to be successful in mitigating the emergence of AMR at farm level.

The socioeconomic preconditions and farm characteristics transcend into the level of disease prevention where farmers in the more peri-urban district Mukono to a higher degree had particular means to protect their animals from disease compared to Lira (73% versus 48%). The reasons for this difference can only be speculated, and a mix of several factors is presumable in accordance with previous research. A study among Ugandan smallholder pig farmers in high-poverty districts highlighted that constraints to animal health were lack of knowledge on husbandry practices and management, poor access to veterinary services, poor hygiene and lack of feed/poor feed quality [29]. Another study, on knowledge, attitudes and practices regarding African Swine Fever (ASF) among smallholder pig farmers in rural Uganda, showed that hurdles for implementing further control measures were lack of knowledge, lack of capital and income, cultural taboo, lack of control of animal movements and lack of regulation enforcement [30]. Worth noting is that in our study we found no differences between the districts among which disease prevention measures were taken. Also, and perhaps worrisome from an AMR perspective, the most common disease prevention measure in both districts was the prophylactic use of veterinary drugs.

One area that closely relates to the possible spread of pathogens and AMR bacteria is the handling of the manure. Discarding it in the environment, leaving it in open air or using it untreated as fertilizer can lead to the spread of pathogens via the environment or food products later consumed by humans or other animals [31,32]. Hence, risk of the spread of disease and AMR through manure was present in both districts, even though the manure handling routines differed. Further, doing nothing to take care of the manure increases the risk for infections among the animals at the farm. It is also worth highlighting that the choices to do nothing with the manure, discard it into the environment or to leave it in open air are to be seen as a waste of resources since, if treated correctly, using it as a fertilizer may enhance crop yields [33,34].

In Lira, disease rates among the pigs, both in long- and short-term, were higher than in Mukono for both respiratory and intestinal diseases as well as sudden death. There are several possible reasons for this, but our results highlight lower access to professional animal health services and lower level of disease prevention as two likely contributors. Regardless of the reasons, our results suggest that the risk for AMR development due to high disease pressure, and subsequent high antibiotic use (rational or irrational), is to be considered more pronounced in Lira than in Mukono. The good access to veterinary drugs further strengthens this reasoning.

Besides affecting the disease rates negatively (e.g., less advice on disease prevention, not getting treatment started in time), the lower access to professional animal health services in Lira reduces the possibilities to handle disease correctly, increasing the risk of inappropriate antibiotic use. However, difficulties in making a correct diagnosis in case of disease exist in both districts, since laboratory testing was practically never included in the animal health services offered. Without a correct diagnosis, there is an obvious risk of inappropriate antibiotic use, which in this regard increases the risk of AMR development in both districts, independently of the access to animal health services. It has previously been shown that, due to the lack of laboratory services, animal health service personnel in these districts often recommend broad-spectrum antibiotics like oxytetracycline to farmers [35], which are known to be prone to induce resistance [36,37].

The difference in access to animal health services is further reflected in the farmers’ responses to pig disease, where farmers in Lira to a larger extent either did nothing or consulted a community animal health worker, while more farmers in Mukono consulted a private veterinarian. The fact that private veterinarians have higher qualifications than community animal health workers suggests that farmers in Mukono to a larger extent get better quality advice when their pigs are sick, which should increase the chance of handling the disease correctly. However, as in many other countries antibiotics can constitute an important source of income for veterinarians, one should therefore be aware that there might be economic incentives for veterinary practitioners to recommend antibiotic treatment to farmers, increasing the risk for over- and misuse [38]. Regarding the income for veterinarians from sales of pharmaceuticals, Dione et al. [35] found antibiotics to be the most profitable drug category in both Lira and Mukono. Favorably, from an AMR perspective, the proportions of farmers that used veterinary drugs in response to disease problems without prior consultation from a professional animal health worker were low in both districts. Besides access to animal health services, how a farmer responds to disease among his/her pigs might obviously also have other reasons, such as economic reasons, cultural reasons or lack of knowledge regarding the disease prevention measures described above.

Based on the above discussion on access to animal health services, it was unexpected that there was no detectable difference between the districts in how many farmers followed treatment instructions from a veterinarian when using veterinary drugs (about half of the famers in both districts). However, in Lira it was almost as common to follow instructions from the pharmacy, whereas this was less the case in Mukono. If the qualification of the person working in the pharmacy is low and the pharmacies’ instructions are followed to a large extent, the risk for non-rational antibiotic use is presumably elevated compared to following instructions from a trained animal health professional.

Even though no difference in access to veterinary drugs was detected between the districts, there was a higher use of antibiotics in pigs in Lira than in Mukono during the four-week period prior to the study, which is to be seen as a risk marker for AMR development. Possibly this is a consequence of the higher disease rates, both long- and short-term, on the farms in Lira compared to Mukono, discussed above. It might also relate to access to, and use of, animal health services, and whose treatment instructions are followed.

Interestingly, despite higher disease rates and more frequent antibiotic use as well as following instructions from pharmacies to a larger extent in Lira than in Mukono, there was no detectable difference in how many farmers had experienced situations where drugs did not work as expected. However, this question deals with drugs in general, and not particularly antibiotics, and also for all animal species at the farm.

A majority of respondents in both districts believed that vaccination prevents disease, which indicates that knowledge levels are good about vaccination in both districts, even though the proportion was larger in Mukono. However, it was more common among the respondents in Lira to believe that vaccination both cures and prevents disease, which might suggest that there are knowledge gaps to be filled.

When it came to knowledge about antibiotics, it was first of all promising from an AMR perspective that the belief that antibiotics are used for fattening was low in both districts. However, some knowledge gaps were also revealed. In Lira, a higher proportion of the respondents than in Mukono believed that antibiotics cure but do not prevent disease, but a higher proportion also believed that antibiotics prevent but do not cure disease (see Table 7). In Mukono, on the other hand, a larger proportion believed that antibiotics both cure and prevent disease. Obviously, it is reasonable to argue that a belief that antibiotics prevent disease increases the risk of using antibiotics for disease prevention, especially if access to antibiotics is good and the knowledge about the risks of AMR is low. As shown above, the most common disease prevention measure in both districts was the use of veterinary drugs, which might be regarded as a red flag, even though we cannot, from this study, say how much of these drugs were antibiotics.

As for all survey studies based on self-reporting, there are some inherent limitations in the precision of this study. These limitations may, for instance, include recall-bias about episodes weeks or months ago. The reliability of some of the data would have been improved by recording observations as well. However, given the design of the study, where data from two districts were compared, we cannot identify any systematic bias in self-reporting between the two districts that may have significantly influenced the comparison and thus the interpretation of the study.

## 4. Materials and Methods

### 4.1. Site Selection

Uganda is a low-income country in East Africa (see Figure 1) with a population of about 47 million people in 2021 [39]. Of all households, in 2014 almost 80% were engaged in agriculture and 69% had subsistence farming as their main income source, although this varied considerably between rural and urban areas (82% versus 29% of households) [40]. On the other hand, only 2% of households had commercial farming as their main income source. The number of agricultural households that kept pigs in 2018 was 1,345,000, with an average of four pigs per farm [41]. The pig population in Uganda is on the rise, with a total of 4.5 million pigs in 2018.

The current study was conducted in Mukono and Lira districts in Uganda (see Figure 1). Mukono district is in central Uganda at 40 km from Kampala, the capital of Uganda, with a human population in 2014 of 596,804 people [40]. In 2008, 63% of all households kept some kind of livestock and 23% kept pigs [42]. The same year, Mukono was one of the districts with the highest number of pigs in the country. Because of the proximity to Kampala, livestock farmers are assumed to have good access to veterinary drugs and animal health services.

Lira district is in Northern Uganda, about 300 km from Kampala with a population of 408,043 in 2014 [40]. In 2008, 80% of all households in the district were engaged in livestock rearing [42] and crop farming and livestock keeping are two main contributors to people’s livelihoods [43]. In 2008, only about 7% of all households kept pigs [42], but pig farming has increasingly become an important enterprise and many people rely on this as their main income source and use the animals as financial assets to pay for school fees and other expenses [43].

### 4.2. Sample Size

The sample size of 240 farms in Lira and 242 in Mukono allowed the detection of a difference between the populations of 0.13 when one of the proportions is 0.5, or 0.1 if one of the proportions is 0.1 or 0.9, at a significance level of 0.05 and with a power of 0.8 (https://epitools.ausvet.com.au/twoproportions, accessed on 20 December 2021).

### 4.3. Farmer Selection

The target population of the study was pig farmers (male and female) in Lira and Mukono districts. District Veterinary Officers (DVO) from both districts were informed prior to the study to identify the top four sub-counties with the highest pig population density to serve as a sampling frame. In each sub-county, two villages were randomly selected for the study, making a total of eight villages per district and 16 villages in total. From each village, 30 farmers were randomly selected to be enrolled in the study. This resulted in a total sample of 240 farmers from Lira and 242 from Mukono.

### 4.4. Drug Profiling

Prior to the farm survey, the investigating team carried out a profiling of all drugs stored in retail shops in urban and peri-urban areas of the respective districts, using a drug profiling tool that has been previously presented [44]. Charts of common drugs were created to facilitate recall of drugs used by farmers during the interview. For farmers who could not recall the drugs they had used in the past months, the drug cards were shown to help them remember.

### 4.5. Data Collection

#### 4.5.1. Questionnaire

The questionnaire used in the study was based on the ‘Antimicrobial use in livestock production systems’ (AMUSE Livestock tool) developed by a working group from the CGIAR Research Program on Livestock (see supplementary material File S2). The questionnaire was developed to facilitate harmonized data collection regarding farmer knowledge, attitudes and practices within the AMR conceptual framework in LMICs, and modified versions of the questionnaire have been used in other African countries, such as Ethiopia [45].

The AMUSE tool comprises 75 questions covering (1) Farm basics and location, (2) Household demographics, (3) Farm characteristics, (4) Management of manure, feed and water, (5) Animal health and disease prevention, (6) Animal health services, (7) Veterinary drug use, and (8) Use of antibiotics. Also, questions about the respondent’s knowledge about the effects of vaccines and antibiotics are included.

#### 4.5.2. Questionnaire Administration

The questionnaire was administered between 13 August and 10 September 2018 and was recorded electronically on tablets using Open Data Kit (ODK) [46], an open-source tool for smart devices (i.e., smartphone or tablet) that enables creation and use of electronic questionnaires. The data collection through ODK was backed up daily.

#### 4.5.3. Training of Enumerators

The field work was led by a research technician who is a veterinarian. In total, eight enumerators were hired for this work. One enumerator was allocated to one sub-country, hence covering two villages. Enumerators were trained on data collection to make sure the interpretations and translations were properly performed and that the use of ODK on the tablets was correct. The questionnaire was then tested by each enumerator on three to five farmers each to make sure that whole data entry procedures worked.

### 4.6. Data Processing and Analysis

Since being a pig farmer was a selection criterion, 17 farms were removed from the dataset after the respondents at these farms stated that they had zero pigs. The selection criterion for the respondent was that the person should play a major role in the management of livestock. Therefore, additionally, two farms were removed as the respondents said that they had no relation to the livestock at the farm. The respondents that did not answer “Management” to the question about their role in relation to livestock answered “Owner” instead (eight in Lira and 17 in Mukono) and were therefore considered enough involved to be kept in the dataset. After the two removals, there were 463 farms left in the dataset, 229 in Lira and 234 in Mukono. In addition, inconsistent, or contradictory, answers were removed.

The processing and the statistical analyses of the data were performed after import to STATA. The two-sample test of proportions was used to compare the two districts regarding categorical data where proportions and samples were large enough to fulfil the assumptions ($n_{S}\cdot \hat{p_{0}}>10$, $n_{S}\cdot \left(1-\hat{p_{0}}\right)>10$, $n_{N}\cdot \hat{p_{0}}>10$ and $n_{N}\cdot \left(1-\hat{p_{0}}\right)>10$ where *n**_N_* and *n**_S_* = sample sizes of the two districts, *p**_N_* and *p**_S_* = proportions in the two districts and $\hat{p_{0}}=\;$($n_{S}\cdot \hat{p_{S}}+n_{N}\cdot \hat{p_{N}})$/($n_{N}+n_{S})$). In the cases that the assumptions were not fulfilled, no statistical tests were performed and “n/a” (not applicable) is noted in the *p*-value column. Mean values were compared using the two-sample *t*-test with equal variances. Both test types were conducted with a significance level of 95%.

## 5. Conclusions

Our findings suggest that knowledge and practices regarding antibiotic resistance and use are not uniform within the same national legal framework but are associated with geographic settings and socioeconomic circumstances. Likely, this variability is larger in countries where the governmental resources to enforce regulations is limited. This in turn calls for AMR-reducing interventions that are tailored for the specific context in which they are to be implemented as a complement to national regulations. To be able to accomplish this tailoring, a systematic collection of knowledge of farm structures, farm practices and access to animal health services and veterinary drugs is necessary.

## Figures and Tables

**Figure 1 antibiotics-11-00251-f001:**
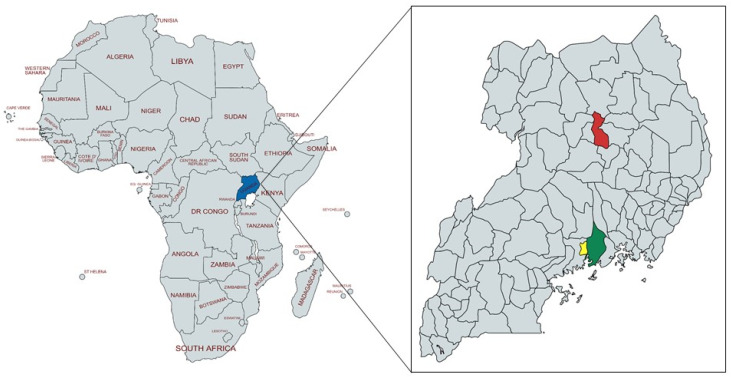
Map of the continent of Africa with Uganda marked in blue (**left**). Map of Uganda with the surveyed districts marked in red (Lira) and green (Mukono), as well as Kampala district marked in yellow (**right**). Source: https://mapchart.net, accessed on 2 December 2021, license: https://creativecommons.org/licenses/by-sa/4.0/ accessed on 2 December 2021.

**Table 1 antibiotics-11-00251-t001:** Comparison of access to animal health services and veterinary drugs in smallholder pig farms in Lira and Mukono districts, Uganda.

Item	Category	Lira	Mukono	*p*-Value
		% (*n*)	% (*n*)	
Does the farm have access to professional animal health services? ^1^	Yes	59.4 (136)	94.0 (220)	0.000
If you have access to animal health services, which ones? (multiple choice) ^2^	State or government	3.7 (5)	1.4 (3)	n/a
	Private full time animal health worker	76.5 (104)	52.7 (116)	0.000
	Both state/government and private	14.7 (20)	30.0 (65)	0.001
	Don’t know	11.8 (16)	16.8 (37)	0.193
If you have access to animal health services, do the animal services include laboratory testing? ^2^	Yes	0.7 (1)	1.8 (4)	n/a
Is the farm involved in a regular animal health service program (all species), e.g., vaccination campaign? ^1^	Yes	45.0 (103)	17.5 (40)	0.000
Do you have access to pharmaceuticals/ veterinary drugs? ^1^	Yes	66.8 (153)	67.9 (159)	0.794

^1 ^Lira *n* = 229 Mukono *n* = 234, ^2^ Lira *n* = 136 Mukono *n* = 220. n/a = not applicable, comparative statistic analysis could not be performed.

**Table 2 antibiotics-11-00251-t002:** Comparison of disease prevention practices in smallholder pig farms in Lira and Mukono districts, Uganda.

Item	Category	Lira	Mukono	*p*-Value
		% (*n*)	% (*n*)	
Do you have any particular means to protect animals from disease (all species)? ^1^	Yes	48.0 (110)	73.1 (171)	0.000
If yes, how (for pigs)? (multiple choice) ^2^	Fencing	14.6 (15)	23.7 (40)	0.070
	Not mixing with other herd/flock	32.0 (33)	40.2 (68)	0.175
	Special feed	3.9 (4)	1.8 (3)	n/a
	Veterinary drugs (incl. vaccines)	55.3 (57)	63.9 (108)	0.161

^1^ Lira *n* = 229 Mukono *n* = 234, ^2^ Lira *n* = 103 Mukono *n* = 169. n/a = not applicable, comparative statistic analysis could not be performed.

**Table 3 antibiotics-11-00251-t003:** Comparison of pig disease response in smallholder pig farms in Lira (*n* = 229) and Mukono (*n* = 234) districts, Uganda.

Item	Category	Lira	Mukono	*p*-Value
		% (*n*)	% (*n*)	
What do you do in response to disease problems (in pigs)? (multiple choice)	Use traditional medicine	0.9 (2)	0.9 (2)	n/a
	Use medicine from veterinary drug store	4.4 (10)	6.0 (14)	0.433
	Consult community animal health worker	34.1 (78)	10.0 (23)	0.000
	Consult private veterinarian	24.9 (57)	65.4 (153)	0.000
	Consult governmental veterinarian	2.6 (6)	6.0 (14)	n/a
	Veterinarian applied/left drugs	0.9 (2)	0.4 (1)	n/a
	I do nothing	35.4 (81)	17.9 (42)	0.000

n/a = not applicable, comparative statistic analysis could not be performed.

**Table 4 antibiotics-11-00251-t004:** Comparison of routines when using veterinary drugs in smallholder pig farms in Lira (*n* = 153) and Mukono (*n* = 159) districts, Uganda.

Item	Category	Lira	Mukono	*p*-Value
		% (*n*)	% (*n*)	
If you have access to veterinary drugs, do you get advice on how to use them?	Yes	73.9 (113)	78.0 (124)	0.393
When using veterinary drugs, whose instructions (kind, dose, length of treatment) do you follow? (multiple choice)	The veterinarian’s	49.0 (75)	55.4 (88)	0.263
	The animal health worker’s	5.9 (9)	15.7 (25)	0.0053
	The pharmacy’s	43.8 (67)	25.2 (40)	0.0005
	Other farmer’s	2.6 (4)	7.6 (12)	n/a
	My own judgement	5.2 (8)	10.7 (17)	n/a
	Other	0.7 (1)	0.6 (1)	n/a
	I don’t get any instructions	3.9 (6)	0 (0)	n/a
	I don’t use veterinary drugs	0.7 (1)	1.9 (3)	n/a
	Don’t know	3.3 (5)	15.1 (24)	0.0003

n/a = not applicable, comparative statistic analysis could not be performed.

**Table 5 antibiotics-11-00251-t005:** Comparison of self-reported antibiotic use in pigs in smallholder pig farms in Lira and Mukono districts, Uganda.

Item	Lira	Mukono	*p*-Value
	% (*n*)	% (*n*)	
Farmers that used at least one antibiotic in pigs the past 4 weeks ^1^	17.9 (41)	7.7 (18)	0.001
Farmers that had an antibiotic as the most commonly used drug in pigs the past 4 weeks ^2^	18.8 (16)	11.8 (9)	0.222

^1^ Lira *n* = 229 Mukono *n* = 234, ^2^ Lira *n* = 85 Mukono *n* = 76.

**Table 6 antibiotics-11-00251-t006:** Comparison of self-reported effect and handling of veterinary drugs in smallholder pig farms in Lira (*n* = 229) and Mukono (*n* = 234) districts, Uganda.

Item	Category	Lira	Mukono	*p*-Value
		% (*n*)	% (*n*)	
Have you experienced situations where drugs did not work (all species)?	Yes (frequently or sometimes)	21.0 (48)	26.1 (61)	0.195
What do you usually do with expired veterinary drugs?	Dispose of	45.9 (105)	37.6 (88)	0.072
	Return to pharmacy	0 (0)	0.9 (2)	n/a
	Give to other farmer	0 (0)	0.4 (1)	n/a
	Use for intended treatment	0 (0)	3.9 (9)	n/a
	Nothing	9.2 (21)	14.5 (34)	0.075
	Never experienced expired drugs	45.0 (103)	42.7 (100)	0.627

n/a = not applicable, comparative statistic analysis could not be performed.

**Table 7 antibiotics-11-00251-t007:** Comparison of knowledge about vaccines and antibiotics among respondents at smallholder pig farms in Lira (*n* = 229) and Mukono (*n* = 234) districts, Uganda.

Item	Category	Lira	Mukono	*p*-Value
		% (*n*)	% (*n*)	
What does vaccination do? (multiple choice)	Cure sick animals	4.4 (10)	3.4 (8)	n/a
	Prevent animals from becoming sick	81.2 (186)	89.7 (210)	0.0092
	Cure sick animals and prevent animals from becoming sick	14.4 (33)	6.8 (16)	0.0081
	Fattening	1.3 (3)	0 (0)	n/a
What do antibiotics do? (multiple choice)	Cure sick animals	89.1 (193)	59.4 (138)	0.000
	Prevent animals from becoming sick	13.1 (25)	3.0 (6)	0.0003
	Cure sick animals and prevent animals from becoming sick	12.7 (34)	37.6 (89)	0.000
	Fattening	3.1 (7)	5.1 (12)	n/a

n/a = not applicable, comparative statistic analysis could not be performed.

## Data Availability

The data has been submitted to the SND public repository at https://hdl.handle.net/20.500.12703/3942 accessed on 2 December 2021.

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
