# Peer review of "Geographic and Socioeconomic Influence on Knowledge and Practices Related to Antimicrobial Resistance among Smallholder Pig Farmers in Uganda"

_antibiotics, 2022, doi:10.3390/antibiotics11020251_

Round 1

Reviewer 1 Report

  1. Line 44: “…handled at global level…” change to “…handled at the global level…”
  2. Line 58: “Some hotspots and emerging hotspots…” change to: “Current and emerging hotspots…”
  3. Line 84: “setting” change to: “settings”
  4. Lines 88-89: The sentence: “In short, we found that legal regulations alone are not sufficient to control antimicrobial use in livestock.” should be moved to chapter Conclusions
  5. Line 106: “as fertilizer” change to “as a fertilizer”
  6. Line 129: Avoid starting the sentence with a number: “73% of the respondents in Mukono said that they had particular means to protect…”. Suggestion: “In Mukono, 73% of the respondents said that they had particular means to protect…”
  7. Line 228: “risk of spread of disease” change to: “risk of disease spread”
  8. Line 233: “as fertilizer” change to “as a fertilizer”
  9. Line 251: “due to lack” change to “due to the lack”

Author Response

Dear Reviewer 1,

Thanks for your valuable comments and suggestions. We have amended the text accordingly in all instances except in two cases as indicated below. 

Line 44: “…handled at global level…” change to “…handled at the global level…”

Line 58: “Some hotspots and emerging hotspots…” change to: “Current and emerging hotspots…”

Line 84: “setting” change to: “settings”

Lines 88-89: The sentence: “In short, we found that legal regulations alone are not sufficient to control antimicrobial use in livestock.” should be moved to chapter Conclusions OUR COMMENT: we think this a matter of style, in some papers it is useful to very briefly indicate the main finding already in the introduction think that's the case here.

Line 106: “as fertilizer” change to “as a fertilizer”

Line 129: Avoid starting the sentence with a number: “73% of the respondents in Mukono said that they had particular means to protect…”. Suggestion: “In Mukono, 73% of the respondents said that they had particular means to protect…”

Line 228: “risk of spread of disease” change to: “risk of disease spread” OUR COMMENT: We think this change will make the sentence cumbersome as the "spread" also includes AMR.

Line 233: “as fertilizer” change to “as a fertilizer”

Line 251: “due to lack” change to “due to the lack”

Reviewer 2 Report

In the manuscript entitled “Geographic and socioeconomic influence on knowledge and practices related to antimicrobial resistance among smallholder pig farmers in Uganda" authrored by Sandra Nohrborg et al., the authors present in an eloquent and clear manner the results of the study aimed to determine the impact of geographic and socioeconomic settings on the antibiotic use in local swine farms in Uganda.

The manuscript does not require editing of English language and style.

The methods are advanced and suitable for the study aim, the results are validaded and statistically analysed.

The importance and novelty of the research are well underlined.

Author Response

Thanks for the positive feed-back!

Reviewer 3 Report

The authors have conducted a quantitative study to identify the application of antimicrobials application in pig farms in Uganda. They chose two geographical locations based on the socioeconomic status of the farmers. It is an excellent study to understand how socioeconomic status influences society. It seems like the farmers have free access to antibiotics without veterinarians’ permission. This study is more beneficial to the local government to implement policies to control the usage of antibiotics, but it might be aware to the general audience.  

In general, it is a well-written manuscript, and the authors have designed the questionnaire well and interpreted it well.

Author Response

Thanks for the positive feed-back!!